# Effect of Precursors on the Electrochemical Properties of Mixed RuOx/MnOx Electrodes Prepared by Thermal Decomposition

**DOI:** 10.3390/ma15217489

**Published:** 2022-10-25

**Authors:** Elisabetta Petrucci, Francesco Porcelli, Monica Orsini, Serena De Santis, Giovanni Sotgiu

**Affiliations:** 1Department of Chemical Engineering Materials Environment, Sapienza University of Rome, Via Eudossiana 18, 00184 Rome, Italy; 2Department of Industrial, Electronic and Mechanical Engineering, Roma Tre University, Via Vito Volterra 62, 00146 Rome, Italy

**Keywords:** mixed-oxide electrodes, thin films, ruthenium oxide, manganese oxide, organic and inorganic precursor, spin coating

## Abstract

Growing thin layers of mixed-metal oxides on titanium supports allows for the preparation of versatile electrodes that can be used in many applications. In this work, electrodes coated with thin films of ruthenium (RuOx) and manganese oxide (MnOx) were fabricated via thermal decomposition of a precursor solution deposited on a titanium substrate by spin coating. In particular, we combined different Ru and Mn precursors, either organic or inorganic, and investigated their influence on the morphology and electrochemical properties of the materials. The tested salts were: Ruthenium(III) acetylacetonate (Ru(acac)_3_), Ruthenium(III) chloride (RuCl_3_·xH_2_O), Manganese(II) nitrate (Mn(NO_3_)_2_·4H_2_O), and Manganese(III) acetylacetonate (Mn(acac)_3_). After fabrication, the films were subjected to different characterization techniques, including scanning electron microscopy (SEM), polarization analysis, open-circuit potential (OCP) measurements, electrochemical impedance spectroscopy (EIS), linear sweep voltammetry (LSV), cyclic voltammetry (CV), and galvanostatic charge–discharge (GCD) experiments. The results indicate that compared to the others, the combination of RuCl_3_ and Mn(acac) produces fewer compact films, which are more susceptible to corrosion, but have outstanding capacitive properties. In particular, this sample exhibits a capacitance of 8.3 mF cm^−2^ and a coulombic efficiency of higher than 90% in the entire range of investigated current densities.

## 1. Introduction

Thin-film electrodes are versatile devices consisting of metal supports—commonly made of titanium or nickel—coated with an electrocatalytic layer of variable thickness ranging from a few nanometers to a few micrometers. Their applications range from energy storage systems [1,2] to sensing [3,4,5,6], photo-electrocatalysis [7], and fuel cells [8]. The possibility of obtaining flexible [9,10,11] and even transparent materials [12] makes these films extremely attractive, even for photo-optoelectronic applications, such as solar cells, diodes, and low-emissivity windows [13]. Except for rare applications as cathodes for the reduction of CO_2_ [14] and oxygen [15], their prominent use in the electrolytic processes is as anodes; they have been extensively explored for wastewater treatments [16,17] to remove common [18] and hazardous pollutants, such as hydrazine [19], bisphenol A [20], and cyanide [21].

Different transition metals and various combinations have been used to prepare the films, which ultimately consist of a thin layer of the corresponding oxide or mixed oxide if more than one metal is used as a precursor.

Thin films can be grown by a multi-step process that involves: (a) the preparation of a solution consisting of the appropriate precursors diluted, in the desired stoichiometric ratio, in hydroalcoholic solvents; (b) the deposition of the solution on the metal support, which can be pristine or previously treated; (c) the solvent evaporation; and, (d) the thermal conversion of the deposit into the corresponding oxide. The thickness of the film can be easily controlled through repeated cycles of deposition and thermal decomposition. The possibilities to customize the properties of thin films by adopting mixtures of precursors or by varying the operating conditions are innumerable. In previous works, we have prepared ruthenium oxide films. In particular, we compared their morphology and electrochemical properties, as well as their tendency to electrogenerate chlorinated inorganic by-products, and their life service by varying the following: i. the metal components (with the addition of non-noble metals) [22,23]; ii. the deposition technique [24] (from casting to spin coating to increase the preparation reproducibility); and iii. the treatment of the metal support (including chemical and electrochemical etching [25,26] and anodization in a fluoride medium to obtain TiO_2_ nanotubes) [27]. A wide range of precursors could be used to obtain RuO_2_ films. However, since Trasatti’s pioneering studies, RuCl_3_ has been the most used chemical [28].

The dependence of electrode performance on the selected precursor has been scarcely investigated. In a study NiO films were grown by the potentiodynamic electrodeposition of nitrate, chloride, and sulfate salts [29]. It has been found that the precursor affected both the morphology and the pseudocapacitance behavior and that sulfate-deriving films exhibited very low impedance and good capacitive behavior due to their honeycomb-like nanostructure. Another study aimed at manufacturing Sb doped SnO_2_ films on Ti by thermal decomposition of SnSO_4_ and SnCl_2_ suggested that SnCl_2_ provided the best results because it implied a larger Sn solubility that resulted in an increase in the electrocatalytic activity of the active film [30]. Regarding the RuOx films obtained by thermal decomposition, ruthenium nitrosyl nitrate (RuNO(NO_3_)_3_) has been proposed as an alternative precursor anion to chloride, mostly with the aim of avoiding residual chlorine. It was proved that the residual chloride entrapped in the RuOx lattice did not affect the performance of the RuOx films. However, to obtain stable crystalline RuOx, nitrate required a higher calcination temperature than chloride [31]. In a subsequent paper, Trasatti concluded that the electrodes obtained from nitrate had a greater surface area, which implied a greater voltammetric area and greater electrocatalytic properties. Other differences might arise from electronic factors related to the size of particles [32]. These findings were confirmed by a later study, in which the dependence of calcination temperature and precursor nature was further investigated by using the same chemicals. RuO_2_ prepared from RuCl_3_ by thermal treatment between 350 °C and 550 °C exhibits constant characteristics in terms of grain size, total area, and voltammetric behavior, unlike ruthenium nitrosyl nitrate, which depends significantly on the temperature [33].

The combination of Ru oxide and Mn oxide has been mainly studied for charge storage in the fabrication of supercapacitors [34,35]. Composite materials endowed with high charge storage capacity and cycling stability were obtained by supporting mixed Mn–Ru oxide on carbon fabric [36] or by dispersing RuOx and MnOx nanoparticles on graphene [37]. Other applications of the mixed oxide include its use as electrodes in Li-ion batteries [38] and in anodic oxidation processes [39].

To improve the stability of the coating, a Ru:Mn ratio of 70:30 is beneficial [40]. However, other studies [23,41] show that by varying the ratios from 70:30 to 50:50, there are no substantial differences either in morphology or corrosion behavior.

Mn(NO_3_)_2_ outperforms other precursors in developing electrocatalytic properties of films when it is used alone [42]. However, the substance exhibits high hygroscopicity, which could cause problems at an industrial scale. In addition, hazardous NO_2_ fumes are released during the thermal decomposition of nitrate. For these reasons, organic precursors have been used in the search for more ecological and easier-to-manage precursors.

The aim of the present work was to establish whether the properties of the thin films of ruthenium and manganese mixed oxide were influenced by the nature of the counter-ion that accompanies the metal ions in the salt precursors. In particular, the focus has been on the use of organic counter-ions, as they usually show better stability and solubility in organic solvents and are more effectively converted in subsequent heat treatments.

In this paper, coatings were obtained via thermal decomposition of different combinations of precursor salts (where Ru and Mn were mixed in a 1:1 molar ratio), which were dissolved in alcoholic solvents. In particular, the adopted salts were: Ruthenium(III) acetylacetonate (Ru(acac)_3_), Ruthenium(III) chloride (RuCl_3_·xH_2_O), Manganese(II) nitrate (Mn(NO_3_)_2_·4H_2_O), and Manganese(III) acetylacetonate (Mn(acac)_3_). The effect of the nature of the precursors on the morphology and electrochemical performance of the fabricated materials was thoroughly characterized. The effect of the treatment of the Ti surface was also included.

## 2. Materials and Methods

### 2.1. Chemicals and Materials

Titanium foils (0.127 mm thick, 99.7% pure), Ruthenium(III) acetylacetonate (Ru(acac)_3_), Ruthenium(III) chloride (RuCl_3_·xH_2_O), Manganese(II) nitrate (Mn(NO_3_)_2_·4H_2_O), and Manganese(III) acetylacetonate (Mn(acac)_3_) were purchased from Merk Life Science in Milano, Italy. Sodium sulphate decahydrate (Na_2_SO_4_·10H_2_O, 99.8%) was supplied from Carlo Erba Reagents in Milano, Italy. All reagents were used without further purification. Double distilled water was used throughout all experiments.

### 2.2. Electrode Preparation

The precursor salts were dissolved in an alcoholic solution in a 0.1 mol L^−1^ concentration and mixed at equal volumes in a 1:1 molar ratio just before being used.

Pristine titanium samples (1.0 × 1.5 cm) were sonicated in a two-step procedure, including a water/acetone 50:50 bath for 10 min and then an ethanol bath for 10 min. Finally, they were rinsed with deionized water and nitrogen dried at an uncontrolled temperature.

The procedure for the preparation of the thin films was as follows: (1) 20 μL of the mixed-precursor solution was poured on the sample. (2) The sample was spin-coated to the speed of 500 rpm for 25 s and held at this value for an additional 20 s. (3) The coated sample was thermally treated in a furnace for 10 min in air atmosphere at 400 °C. The entire treatment was repeated three times. After the last cycle, the electrode was annealed for an additional hour in air atmosphere at 400 °C.

Table 1 shows the electrodes prepared as a function of the nominal composition.

### 2.3. Electrode Characterization

Morphological observations of the surfaces were conducted using an electron microscope ZEISS SIGMA 300 equipped with a GEMINI column (Jena, Germany), and Bruker EDS (Bruker Italia, Milano, Italy).

All the electrochemical measures were performed in 0.1 mol L^−1^ Na_2_SO_4_ in a three-electrode cell where the ultra-thin films were used as working electrodes of 1 cm^2^ area. The Autolab PGSTAT204 integrated with the frequency response analyzer FRA32M was used. The counter electrode was a platinum plate of the same area, while the reference was a saturated Ag/AgCl electrode. During the experiments, the temperature was controlled in the range of 22 ± 1 °C.

The cyclic voltammetry measurements were repeated at least three times and were carried out in the scan rate range between 5 and 500 mV s^−1^, while the linear sweep voltammograms (LSV) were recorded at a scan rate of 5 mV s^−1^ in the 0.5–1.4 V potential range.

Electrochemical impedance spectroscopy (EIS) was recorded in the frequency range from 10 kHz to 10 mHz with an ac amplitude of ±10 mV at the potential of 0.0 Volt vs. reference. The ZView fitting program (Scribner, NE, USA) was used to analyze the EIS spectra through the application of equivalent electrical circuits. The measurements were repeated at least three times to ascertain their reproducibility.

The corrosion properties were tested by potentiodynamic polarization sweeps (from −0.3 to +0.5 V with a scan rate of 2 mV s^−1^) preceded by an open circuit potential (OCP) measurement extended for 1 h.

Linear polarization curves were recorded in the potential range of 0.0–1.4 V at the scan rate value of 2 mV s^−1^.

Galvanostatic Charge–Discharge (GCD) measurements were carried out in the Na_2_SO_4_ 0.1 mol L^−1^ aqueous solution at different current ranges in the potential range from 0–0.9 V vs. Ag/AgCl reference at different current densities.

## 3. Results

### 3.1. Surface Morphology

The SEM images of the prepared electrodes (Figure 1) reveal that their morphology is affected by the adopted combination of precursors.

The samples prepared from the solutions containing at least one inorganic precursor—E-01 (Figure 1a) and E-04 (Figure 1c)—partially retain the typical appearance of the titanium commercial foils, indicating that the coatings are extremely thin. The thickness, though not measured, can be estimated at a few nanometers, while the amount of the active material was of the order of a few tenths of milligrams. Their EDS maps (Appendix A) also show a prevalence of titanium with a good distribution of ruthenium and manganese. However, on the E-04 sample, it is possible to observe some islets denoting thicker areas, which were probably created by the uneven dispersion of the precursor solution in the spin-coating phase. Corresponding to these points, the underlying titanium appears to be completely covered.

The use of exclusively organic precursors promotes the growth of thicker films with more effective coverage of the underlying titanium, whose surface texture and specific signal in the EDS maps are less evident. This behavior can be presumably attributed to the higher viscosity of the organic solutions that adhere better to the surface of the material when spin coated. By comparing electrodes E-02 (Figure 1b) and E-03 (Figure 1d), it can be observed that the etching pretreatment impairs the combination of ruthenium and manganese with irregular accumulation on the crests of the etched areas. Therefore, considering the coating homogeneity and distribution, the untreated sample prepared from organic precursors (E-02 sample) appears to be the most promising.

### 3.2. Corrosion Tests

The OCP plot in Figure 2a illustrates the trend of the equilibrium potential in the first 60 min. As can be seen, the electrodes containing an inorganic precursor (E-01 and E-04) are readily stabilized. Their curves immediately reach a steady-state value, which remains unchanged for the entire measurement time. On the other hand, the electrodes prepared with organic precursors (E-02 and E-03) show a notable drift, with a decreasing slope that denotes their difficulty in reaching stabilization, especially in the first half hour of measurement. Since these last two samples behave very similarly, it can be deduced that the time needed to achieve the steady state is not affected by the modification of the surface, but rather by the composition of the coating. The potential value reached by each electrode after 60 min is reported in Table 1.

Figure 2b shows the potentiodynamic polarization curves in a 0.1 M Na_2_SO_4_ solution. These tests provide useful information on susceptibility to corrosion. The corrosion parameters E_corr_ and J_o_ (reported in Table 1) are obtained from the graph respectively as the abscissa and ordinate of the intersection of the tangent at the cathode branch with the tangent at the anode branch.

The values of the corrosion potentials of the electrodes, as well as the current values, are all included in narrow ranges from −96 mV to 24 mV and from 1.47 mA to 6.55 mA, respectively.

However, the E-01 electrode, containing RuCl_3_ as a precursor, differs most from the others, as it displays poorer corrosion resistance (more negative potential and higher current density). The worst performance of this electrode might depend on the destabilization caused by chloride residues deriving from the ruthenium precursor and the resulting redox phenomena.

For E-01, the slope of the cathode branch, where HER is the dominant cathodic reaction, is markedly greater, denoting the occurrence of some reduction process related to the ability to acquire electrons, probably due to the manganese atoms present on the surface with different oxidation states. On the contrary, in all the anodic branches, the current quickly reaches a plateau, corresponding to the passivation conditions, where corrosion proceeds uniformly.

In the anodic branch of the etched electrode, E-03, the presence of a shoulder suggests the presence of a second anodic process, probably due to different regrowth of the oxide film after the HF treatment in the titanium surface layer [43,44].

### 3.3. Voltammetric Tests

Figure 3 reports the CV curves of the four electrodes at different scan rates (from 5 to 500 mV s^−1^) in the potential range of 0.0–0.9 V. The choice to explore a narrow range of values is dictated by the intent to investigate the effect of precursors on capacitive or pseudocapacitive behavior, since these coatings are often used in energy storage applications. The E-01 electrode (Figure 1a) displays current values at least double compared to the others, thus differing from a capacitive point of view. All the electrodes present pseudo-rectangular CV curves, indicating quasi-ideal electric double-layer capacitive behavior. This behavior also persists at higher scan rates for all films except E-01. For the latter, elongated CV at the fastest scan is observed, probably due to a less efficient ion migration to the electrodes caused by reduced ion diffusion and transport. This behavior is essentially observed in the presence of porous or mesoporous films [45], thus suggesting that the E-01 film is presumably endowed with greater surface development.

The trends can be better visualized by considering the values of the voltammetric charge (Figure 4), q*, extracted from the CV curves considering only the positive current at the different scan rates, as previously reported [23]. The E-01 electrode outperforms the others, which show very similar values and trends. This behavior seems to indicate that the charge seems unaffected by both the surface modification and the nature of the manganese precursor.

The catalytic performance of the films is investigated by linear sweep voltammetry (LSV) in the potential range of 0–1.4 V. The curves in Figure 5 indicate that the electrodes prepared with organic precursors (E-02) (E-03) exhibit lower oxygen evolution reaction (OER) activity. In particular, the two curves show practically identical capacitive behavior with significant differences in correspondence with the solvent discharge. The onset values are similar and equal to 1.131 and 1.170 V for smooth (E-02) and etched (E-03) samples, respectively. However, the etched sample shows a more rapid increase in the current, probably due to the greater surface area resulting from the treatment undergone before the film deposition. The overpotential for oxygen evolution is reduced by the replacement of the organic precursor of manganese with Mn(NO_3_)_2_ (E-04), and even more so by the replacement of the organic precursor of ruthenium with RuCl_3_ (E-01). The difference found in the adoption of the different precursors can presumably be attributed both to a different balance of the oxidative states of ruthenium and manganese [46], and to an alteration of the energy values of interaction between the active sites and the oxygen-containing intermediates, which can delay or anticipate the oxygen evolution [47].

### 3.4. EIS Analysis

To further investigate the kinetic properties of the films, electrochemical impedance spectroscopic (EIS) measurements were performed.

Spectra were interpreted using an equivalent electrical circuit model (Figure 6a, inset) consisting of a resistance (R_Ω_) in series with a parallel circuit of charge transfer resistance (R_ct_) and constant phase element (CPE).

Fitting results are shown in Table 2.

Effective capacitance was calculated from EIS data according to the equation:(1)C=(Q0Rct)1nRct

The electrode E-01 offers a low-resistance path for charge transport, resulting in appreciable capacitance values, especially considering the very thin oxide film realized on the sample. Despite the difference in surface morphology, the etching procedure does not seem to influence the E-02 and E-03 kinetics, as corresponding fitting parameters were obtained for both electrodes. These specimens also possess a higher resistance to corrosion, thus confirming what has been found in the potentiodynamic tests, at the expense of their capacitive ability.

As E-04 exhibits intermediate behavior, it might be assumed that the presence of an inorganic component among the precursor is significant for increasing the capacitance of the materials, with ruthenium playing a key role.

The higher values of the n parameter indicate that the samples E-02 and E-03 present the most homogeneous charge distribution, suggesting that the organic precursors could promote the formation of a more even coating. However, given the slight differences with the other electrodes, this specific point needs further investigation.

### 3.5. Galvanostatic Charge–Discharge Tests

Figure 7a illustrates the galvanostatic charge–discharge curves of the four considered films recorded at a current density value of 500 μA cm^−2^. All materials present rather symmetric charge–discharge profiles, thus indicating a potential for capacitive applications. However, the E-01 sample displays an extended charge–discharge time, and therefore a better capacitive performance than the other three films, which behave quite similarly. These data agree with what has been previously observed in the CV and EIS analyses.

The areal-specific capacitances (*C_s_*) of the prepared films from the cyclic voltammograms in Figure 3 were calculated using the following equation:(2)$$C_{s}=\frac{1}{Av\left(V_{2}-V_{1}\right)}\int_{V_{1}}^{V_{2}}I\left(V\right)dV$$
where *A* represents the surface of the electrode (cm^2^), *v* represents the scan rate value (V s^−1^), the difference (*V*_2_ – *V*_1_) represents the potential window (*V*), and *I* represents the current (*A*) [48].

The *C_s_* value found for the E-01 electrode is 8.3 mF cm^−2^, while the other films have values of about 1.2 mF cm^−2^. Considering the extreme thinness of the coating, the capacitance behavior of E-01 is particularly promising. Because of its performance, GCD curves for the E-01 sample have been repeated at different current densities in the range of 50–1000 μA cm^−2^ (Figure 7b). As can be seen, higher scan rates worsen the capacitive behavior, probably due to the reduced diffusion of the electrolyte [48]. Nonetheless, the curves maintain their triangular shape in the entire range of investigated currents and, additionally, show good cycle stability (Figure 7c).

Finally, we calculated the Coulombic Efficiency (CE), which describes the charge efficiency by which electrons are transferred during charging–discharging cycles [49]:(3)$$CE=\frac{t_{D}}{t_{C}}\cdot 100$$

In Equation (3), *t_D_* and *t_C_* represent the discharging and charging time (s).

The CE values are higher than 70% for all the electrodes (Figure 7a), except for E-01, which exhibits values higher than 90% in the whole current density range (Figure 7b), thus indicating its superior stability.

## 4. Conclusions

Ultra-thin films of mixed RuOx and MnOx were prepared via spin coating. The effect of the type of precursors was investigated through morphological and electrochemical tests.

The results reveal that the adoption of different precursors affects the electrochemical response of the films by differentiating the structuring of the coating in terms of compactness, uniformity, and, indirectly, surface development. However, further studies are needed to confirm this correlation, particularly on thicker films.

From the experimental tests, the following conclusions can be drawn:-The Ru inorganic precursor (E-01) promotes the growth of a thinner and less compact film with a larger specific surface area. This provides a higher amount of electrochemically active sites for OER and an enhancement of the capacitive behavior with a more remarkable tendency for charge storage. The replacement of Mn(NO_3_)_2_ with Mn(acac)_3_ does not affect the viability of the process as the price of the two salts are quite comparable-The replacement of RuCl_3_ with organic precursors improves the corrosion resistance and shifts the oxidation potential of the aqueous electrolyte to a more positive potential, thus promoting the processes based on chlorine evolution.-The electrodes obtained from only organic precursors (E-02 and E-03) exhibit thicker coatings that suffer from slow OCP stabilization but present the best charge distribution.

## Figures and Tables

**Figure 1 materials-15-07489-f001:**
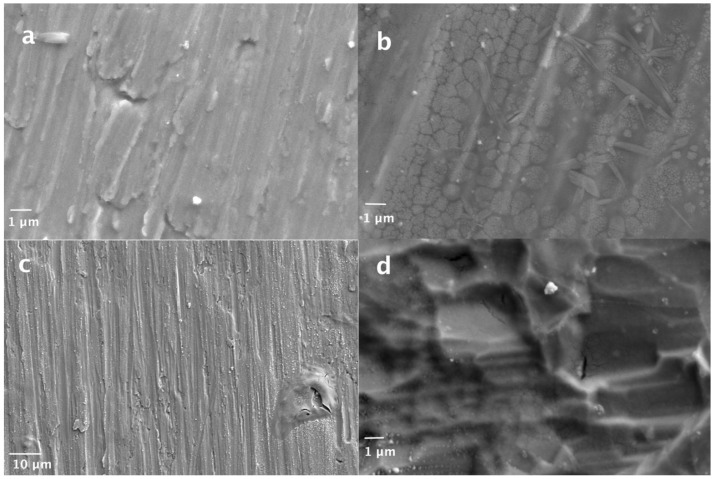
SEM images of: (**a**) electrode E-01 (precursors: RuCl_3_ and Mn(acac)_3_; smooth); (**b**) electrode E-02 (precursors: Ru(acac)_3_ and Mn(acac)_3_; smooth); (**c**) electrode E-04 (precursors: Ru(acac)_3_ and Mn(NO_3_)_2_; smooth); (**d**) electrode E-03 (precursors: Ru(acac)_3_ and Mn(acac)_3_; etched).

**Figure 2 materials-15-07489-f002:**
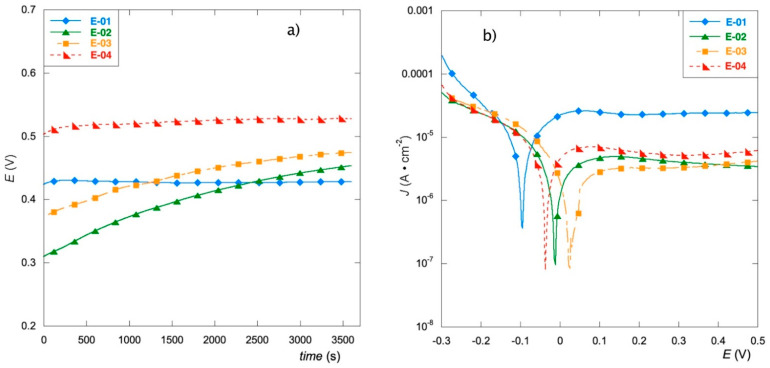
OCP values (**a**) and polarization curves (**b**) of the prepared materials in Na_2_SO_4_ 0.1 mol L^−1^ solution.

**Figure 3 materials-15-07489-f003:**
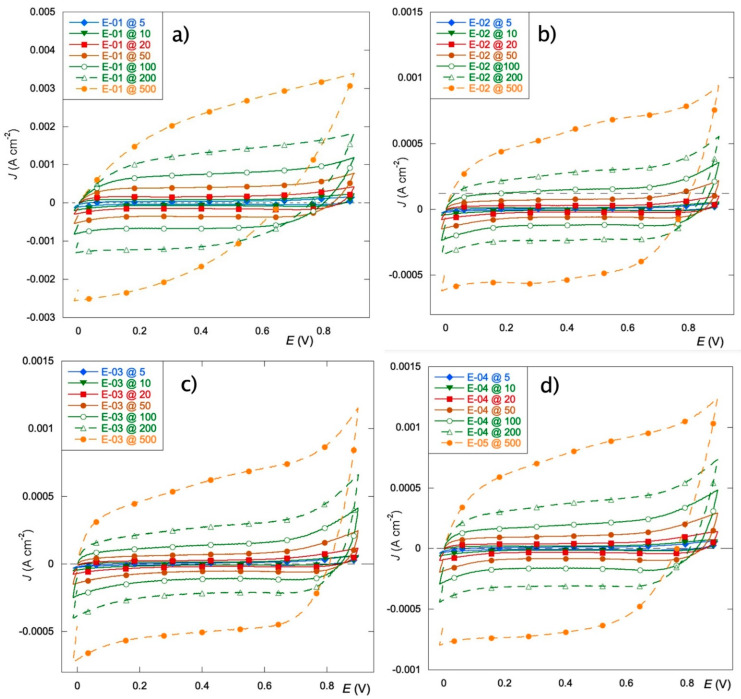
Cyclic voltammetries at scan rates in the range 5–500 mV s^−1^ in Na_2_SO_4_ 0.1 mol L^−1^ for: E-01 (**a**), E-02 (**b**), E-03 (**c**), E-04 (**d**).

**Figure 4 materials-15-07489-f004:**
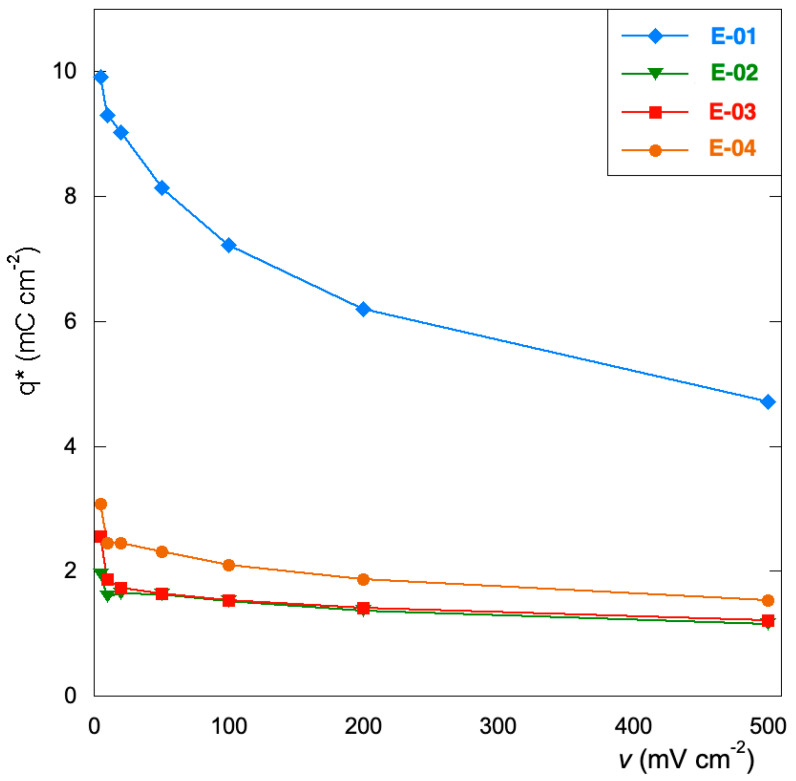
Voltammetric charge (q*) as a function of the scan rate values.

**Figure 5 materials-15-07489-f005:**
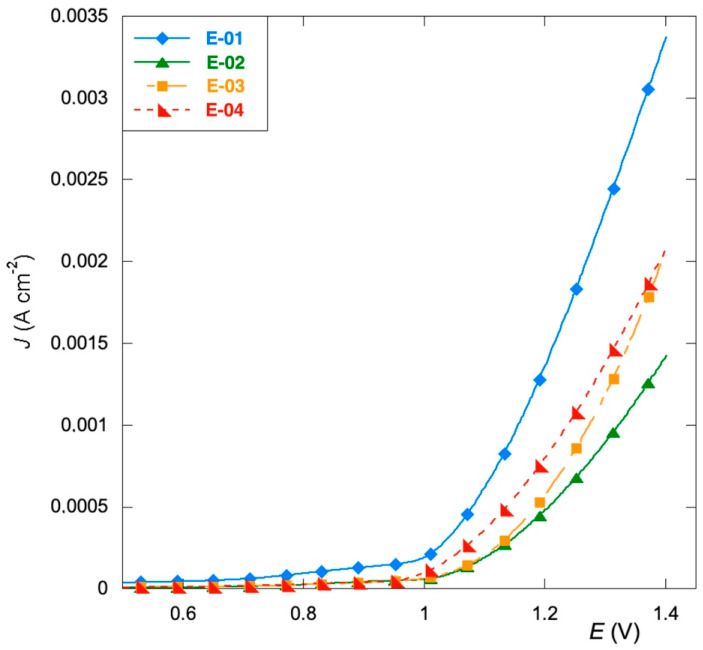
Linear sweep voltammetry recorded at 5 mV s^−1^ in Na_2_SO_4_ 0.1 mol L^−1^.

**Figure 6 materials-15-07489-f006:**
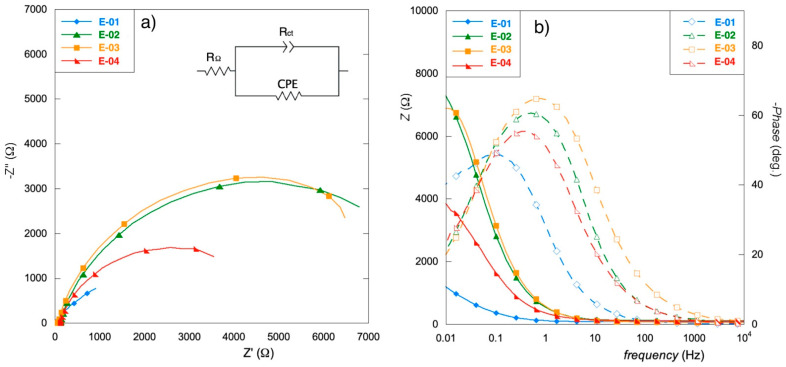
Nyquist (**a**) and Bode (**b**) diagrams of EIS spectra.

**Figure 7 materials-15-07489-f007:**
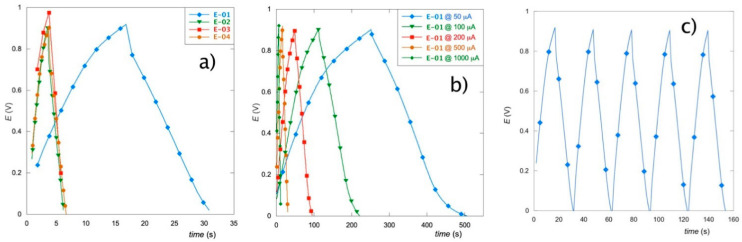
Galvanostatic charge–discharge curves of all the prepared electrodes at 500 μA cm^−2^ (**a**); of E-01 sample in the 50–1000 μA cm^−2^ current density range (**b**); Cycle stability of E-01 sample at 500 μA cm^−2^ (**c**). Potential range: 0.0–0.9 V. Electrolyte: Na_2_SO_4_ 0.1 mol L^−1^.

**Table 1 materials-15-07489-t001:** Electrode characteristics: composition and corrosion data.

Item	Surface Finish	RuCl_3_	Ru(acac)_3_	Mn(NO_3_)_2_	Mn(acac)_3_	OCP (V)	E_corr_ (V)	J_0_ (mA cm^−2^)
E-01	smooth	X			X	0.428	−0.096	6.55
E-02	smooth		X		X	0.453	−0.013	2.02
E-03	etched		X		X	0.474	0.024	1.47
E-04	smooth		X	X		0.528	−0.036	3.06

**Table 2 materials-15-07489-t002:** Fitting results of the EIS spectra in Figure 3.

	E-01	E-02	E-03	E-04
R_Ω_ (Ω)	81.1	117.0	95.0	107.0
R_CT_ (kΩ)	2.3	8.4	8.6	4.9
Q_0_ (mMho s^n^)	4.30	0.45	0.40	0.78
n	0.74	0.81	0.81	0.76
C (mF)	9.43	0.61	0.53	1.18

## Data Availability

The data presented in this study are available on request from the corresponding author.

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
