# Peer review of "Effect of Precursors on the Electrochemical Properties of Mixed RuOx/MnOx Electrodes Prepared by Thermal Decomposition"

_materials, 2022, doi:10.3390/ma15217489_

Round 1
Reviewer 1 Report
The authors studied the effect of precursors on the electrochemical properties of mixed metallic oxide electrodes prepared by thermal decomposition. The property of the obtained materials is pretty good, and the characterizations can support their conclusions. So I recommended its publication in Materials after revisions as follows:
1. This study objects are Ru and Mn-based oxides, so in the introduction parts, the authors should summary more advances on Ru and Mn-based oxides or their composites on similar studies.
2. Why are the Mn-based reagents chosen to modify Ru-based materials?
3. How is the mixed ratio of 1:1 chosen? Is that ratio optimal for the electrochemical property?
4. The control experiment of using mixed RuCl3 and Mn(NO3)2 as precursors should be added to better know the advantages of using organic precursors
5. In-depth discussion can be added in the conclusion part about the underlying mechanism of improving properties by using inorganic/organic mixed precursors
6. Can the high-performance materials or effective strategies be used for development of flexible electronics which is a frontier and hotspot at present? An outlook in Conclusion may help promote its further development. Here some closely related works are provided for your references: 1) DOI: 10.1016/j.matt.2021.07.021; 2) DOI10.1002/admi.201500711; 3) DOI: 10.1007/s42765-022-00162-7
7. There are some grammar errors in the manuscript, such as, line 30, “tor nickel”; line 38 “such…”. The authors should check the manuscript carefully and correct all of them.
Reviewer 2 Report
Petrucci et. al. designed the electrodes coated with a thin film of ruthenium (RuOx) and manganese oxide (MnOx) using the thermal decomposition of precursor solution methods and measured the CV and GCD. The obtained value of specific capacitance was 8.3 mF. The following issues need to be further addressed before the consideration of publication in Materials;
1) X-ray diffraction is the basic technique to study the composition and material properties, such as crystallinity and amorphous nature. So, I suggest the authors should provide the XRD data
2) The authors mentioned that the precursors change the surface morphology and surface area. The specific capacitance is related to the surface area, the authors should provide the BET measurement.
3) Why did the authors not measure the thickness of films, it is necessary to estimate the specific capacitance?
4) What about the specific capacity of the device?
5) The authors should also measure the energy and power density of the device
6) What about the stability of the device, the authors should measure the device up to 10000 cycles and measure the charging/discharging and efficiency of the device
7) The authors should improve the abstract of the manuscript, most of the part is related to the materials and methods, not about the outcomes.
Reviewer 3 Report
Authors have fabricated the ruthenium and manganese oxide thin film coated electrodes using thermal decomposition of a precursor solution. Authors have studied the effect of different precursors on the various properties of the material. The manuscript is well written but lacks in certain areas. Therefore, the manuscript needs a major revision before further publication process. The following point must be considered during the revision of the manuscript:
Major concern:
1. Authors considered the smooth surface of thin films but the results contradict the statement as non-uniform and different structures are visible in the provided SEM images. Provide high-resolution SEM images at lower scales for different samples. Also, the visibility of the inset figures is poor.
2. Authors stated the material’s high specific surface area but didn’t provide any characterization related to it. Provide the specific surface area results in the manuscript for better understanding to readers.
3. Provide a schematic for the experimental process to better understand the work performed.
Minor Issues:
4. Some key parameters are missed in the experimental section. Provide step size for different characterizations in the experimental section.
5. Provide figures having at least 300 dpi resolution.
6. All the references are not as per the journal guidelines.
Round 2
Reviewer 1 Report
The authors have made revisions according to the comments, but further improvements are still needed. Due to its defect in experimental design, its publication should be reconsidered. The revision suggestions are as follows:
1. For a research paper, its experimental design must be complete and rigorous. While the experimental design in this paper is defective. For example, the mixed ratio of Ru/Mn is chosen as 1:1 casually without scientific basis. That greatly decrease the scientificity of this study and credibility of part conclusions. Although the authors gave their explaining for this, but that can’t hide the flaws in the design.
2. Any study or experimental design has its scientific logic. This study seems blind in that aspect. So I suggest the authors expound the underlying reason for the design of using Mn to mix with Ru in the text of the manuscript.
3. The authors wrote that in the response letter that “However, because of its instability due to the high hygroscopicity, it is difficult to prepare mixed precursors' solutions with well-established stoichiometric ratios.” Through a rational experimental setup, it is not difficult to avoid its hygroscopicity.
4. The important references shouldn’t be ignored compared to the one in the manuscript: 10.1016/j.matt.2021.07.021; 10.1007/s42765-022-00162-7
Reviewer 2 Report
Accept in the present form
Author Response
The authors would like to thank the reviewers for their helpful comments and suggestions, following which the manuscript has been revised.
Response to Reviewer #2
Accept in the present form
- We sincerely thank the Reviewer.
Reviewer 3 Report
The authors have provided a response to a maximum number of comments in the revised manuscript. This manuscript can be accepted after mandatory minor revision. The following point must be considered during the revision of the manuscript:
Minor concern:
1. In figure 1, the visibility of the inset is not as per the Journal standards. Either improve the quality of inset figures or represent the elemental mapping figures in another figure for better visibility to readers.
